# Identification and Characterization of MmuPV1 Causing Papillomatosis Outbreak in an Animal Research Facility

**DOI:** 10.3390/v17091204

**Published:** 2025-09-01

**Authors:** Vladimir Majerciak, Kristin E. Killoran, Lulu Yu, Deanna Gotte, Elijah Edmondson, Matthew W. Breed, Renee E. King, Melody E. Roelke-Parker, Paul F. Lambert, Joshua A. Kramer, Zhi-Ming Zheng

**Affiliations:** 1Tumor Virus RNA Biology Section, HIV Dynamics and Replication Program, Center for Cancer Research, National Cancer Institute, National Institutes of Health, Frederick, MD 21702, USA; vladimir.majerciak@nih.gov (V.M.); lulu.yu@nih.gov (L.Y.);; 2Laboratory Animal Medicine, Laboratory Animal Sciences Program, Frederick National Laboratory for Cancer Research, Bethesda, MD 20892, USA; kristin.killoran@nih.gov (K.E.K.); mbreed@clemson.edu (M.W.B.); melody.roelke-parker@nih.gov (M.E.R.-P.); josh.kramer@nih.gov (J.A.K.); 3Molecular Histopathology Laboratory, Laboratory Animal Sciences Program, Frederick National Laboratory for Cancer Research, Frederick, MD 21701, USA; elijah.edmondson@nih.gov; 4McArdle Laboratory for Cancer Research, University of Wisconsin, Madison, WI 53705-2275, USA; renee.king@wisc.edu (R.E.K.); plambert@wisc.edu (P.F.L.)

**Keywords:** MmuPV1, papillomavirus, laboratory animal infection, PCR, viral genome

## Abstract

Mouse papillomavirus (MmuPV1) is the first papillomavirus known to infect laboratory mice, making it an irreplaceable tool for research on papillomaviruses. Despite wide use, standardized techniques for conducting MmuPV1 animal research are lacking. In this report, we describe an unexpected MmuPV1 outbreak causing recurrent papillomatosis in a specific pathogen-free animal research facility. The infected mice displayed characteristic papillomatosis lesions from the muzzles, tails, and feet with histological signs including anisocytosis, epithelial dysplasia, and typical koilocytosis. Etiology studies showed that the papilloma tissues exhibited MmuPV1 infection with expression of viral early and late genes detected by RNA-ISH using MmuPV1 antisense probe to viral E6E7 region and antisense probe to viral L1 region. The viral L1 protein was detected by an anti-MmuPV1 L1 antibody. PCR amplification and cloning of the entire viral genome showed that the origin of the outbreak virus, named MmuPV1 Bethesda strain (GenBank Acc. No. PX123224), could be traced to the MmuPV1 virus previously used in studies at the same facility. Our data indicate that MmuPV1 could exist in a contaminated environment for a long period of time, and a standardized international animal protocol discussing how to handle MmuPV1 studies is urgently needed.

## 1. Introduction

Papillomaviruses are small non-enveloped, double-stranded circular DNA viruses and infect numerous species of animals, from fish to humans [1]. This group of viruses is replicated in host mucosal and cutaneous epithelia, often without any pathological manifestation. However, in some cases, the papillomavirus infection leads to the formation of benign (warts) or malignant tumors [2,3]. The tumor formation is often associated with viral genome integration into the host genome. Although virus genome integration is a dead end for viral genome replication and virus production, the integrated viral DNA sustains increased expression of viral oncogenes E6 and E7, which are essential for cell immortalization and proliferation. The virus genome integration is random and occurs mostly in the intergenic or noncoding regions [4], with multiple integration sites commonly seen in high-risk HPV tumors. However, recent discoveries indicate that only one of multiple integrated HPV DNA copies is selected to express E6 and E7 oncogenes for cell proliferation and clonal cell expansion [4,5].

Infections with papillomaviruses are species- and tissue-specific, and, thus, no animals can be infected with any human papillomavirus. However, given the wide range of species-specific viruses, other animal papillomaviruses have been used as HPV surrogates. Historically, bovine papillomavirus (BPV) cow infection and cottontail rabbit papillomavirus (CRPV) rabbit infection were two animal models used to understand papillomavirus infection and pathogenesis [6,7]. However, these animal models are costly and genetically heterogenous, allowing only limited experimental intervention. Mouse papillomavirus type 1 (MmuPV1), originally designated as MusPV, is the first know papillomavirus which infects the laboratory stain of mice (*Mus musculus*) [8], thus providing an invaluable tool for papillomavirus research [9,10]. In infected animals, MmuPV1 exhibits broad infection tropism of both mucosal and cutaneous epithelia, independently of host immune status. These features of MmuPV1 allow laboratories to study the natural history of papillomavirus infection, transmission, immune responses and carcinogenesis in a well-studied and genetically manipulatable host [11,12,13].

To date, numerous studies have been conducted using MmuPV1 infection in different experimental settings. In this report, we describe an unanticipated recurrent outbreak of MmuPV1 infection in an animal facility three years after the facility was used for MmuPV1 research. We show the widespread and long persistence of infectious MmuPV1 throughout the animal facility and multiple unintended infections of susceptible animals [14]. Molecular analysis of the virus, named MmuPV1 Bethesda strain, had its origin tracing to the synthetic MmuPV1 virus used in the initial study in the facility. Our careful investigation concludes that an established international protocol and regulations on MmuPV1 virus safety and animal management are urgently needed.

## 2. Materials and Methods

### 2.1. Ethical Statement

All animals were maintained in accordance with the Guide for the Care and Use of Laboratory Animals in a facility accredited by AAALAC International. Animal use was conducted humanely and approved by the National Cancer Institute-Bethesda’s Animal Care and Use Committee (NCI-ACUC) on an animal protocol ASB-001 titled “Infectious Disease Monitoring in NCI Rodent Facilities” that was approved on 4 August 2021.

### 2.2. RNA In Situ Hybridization (RNA-ISH)

A detailed description of RNA-ISH using RNAscope technology (Advanced Cell Diagnostics or ACD, Newark, CA, USA) can be found in our previous publication [4,13]. Mouse tissues were fixed using 10% Neutral-Buffered Formalin, embedded in paraffin, and cut into 5 μm sections. Viral genomic DNA was removed by DNase I digestion (Thermo Fisher Scientific, cat. # EN0521, Frederick, MD, USA) followed by RNAscope Protease Plus treatment. RNA in situ hybridization (RNA-ISH) was performed with RNAscope Multiplex Fluorescent Detection Reagents V2 (ACD, cat. # 323110) for florescence dual staining. MmuPV1 E6E7 (ACD, cat. # 409771-C2) antisense probe in channel 2 was mixed with an MmuPV1 L1 antisense probe (ACD, cat. # 473278) in channel 1 before hybridization. The signal was detected using tyramide signal amplification the (TSA) Fluorescein Evaluation Kit (Perkin Elmer, cat. # NEL760001KT, Shelton, CT, USA). Nuclei were counterstained with DAPI. The mounted slides were scanned at 20× resolution using an Aperio CS2 Digital Pathology Scanner (Leica Biosystems, Deer Park, IL, USA) together with representative hematoxylin and eosin (H&E)-stained section.

### 2.3. Indirect Fluorescent Antibody (Ifa) Staining

Serial Formalin-Fixed Paraffin-Embedded (FFPE) tissue sections were dually stained for MmuPV1 L1 capsid protein and K14 using the TSA system, as previously described [13]. Before staining, the slides were deparaffinized, rehydrated, and quenched with 3% H_2_O_2_ in methanol for 10 min. The antigen was retrieved with Tris-based pH 9.0 buffer (Vector Laboratories, cat. # H-3301, Newark, CA, USA). Sections were permeabilized in 0.1% PBS-Tween and blocked in Tris-NaCl-blocking (TNB) buffer (Perkin Elmer cat. # FP1020). Slides were first incubated with the rabbit anti-L1 primary antibody (1:5000; gift from Dr. Chris Buck, National Cancer Institute, National Institutes of Health) at 4 °C overnight followed by incubation with anti-rabbit horseradish peroxidase (1:500; Jackson ImmunoResearch, cat. # 111-035-144, West Grove, PA, USA). The washed slides were then incubated with biotin-tyramide TSA solution (1 mg/mL) diluted 1:500 in reaction buffer (0.001% H_2_O_2_ in 0.1 M imidazole). The L1-stained slides were then incubated with rabbit anti-K14 primary antibody (1:1000; BioLegend, cat. # 905301, San Diego, CA, USA). Finally, the slides were incubated with goat anti-rabbit Alexa Fluor 488 (1:200; Invitrogen, cat. # A-11008) and streptavidin-conjugated Alexa Fluor 594 (1:200; Invitrogen cat. # S-32356, Carlsbad, CA, USA). Nuclei were counterstained with Hoechst 33342 (Thermo Fisher Scientific, cat. # H3570) and mounted in Prolong Diamond mounting media (Thermo Fisher Scientific, cat. # P36970). The slides were scanned with Aperio CS2 Digital Pathology Scanner (Leica Biosystems) and processed by ImageJ software (accessed on 28 April 2022, https://imagej.net/ij/).

### 2.4. Total DNA Isolation

Total DNA from papilloma tissues was extracted using Quick-DNA Miniprep Plus Kit (Zymo Research, cat. # D4068, Irvine, CA, USA) according to manufacturers’ protocol. Papilloma tissues (~10 mg) were cut into small pieces and treated with proteinase K for 5 h at 55 °C while shaking at 300 rpm. The undigested debris was removed via centrifugation at 12,000× *g*. The supernatant was mixed with Genomic Binding Buffer and transferred into a Zymo-Spin IIC-XLR column. The column-bound DNA was washed once with DNA Pre-wash buffer and twice with DNA Wash buffer. DNA was eluted in DNA elution buffer, and DNA concentration was determined by NanoDrop (Thermo Fisher Scientific).

### 2.5. Polymerase Chain Reaction (PCR)

PCR amplification was performed in reactions containing 1 × Platinum SuperFi II Green PCR Master Mix (Thermo Fisher Scientific, cat. # 12369010), 100 pmol of primers and ~250 ng of total genomic DNA using 35 amplification cycles with 2–10 min extension time. The primers used for amplification are summarized in Table 1. The obtained PCR products were analyzed via agarose gel electrophoresis, purified by Zymoclean Gel DNA Recovery Kit (Zymo Research, cat. # 11-301C) and subjected to Sanger sequencing.

### 2.6. Genome Sequence Alignment

The original genome sequences were modified such that the first position corresponds to the first nucleotide of E6 open reading frame (ORF). The modified sequences were aligned by Clustal Omega (accessed on 25 June 2025, https://www.ebi.ac.uk/jdispatcher/msa/clustalo).

### 2.7. Cloning of the MmuPV1 Bethesda Strain Genome

The full-length MmuPV1 Bethesda strain genome was PCR-amplified from total DNA isolated from papilloma tissues using Platinum SuperFi II Taq polymerase and paired primers summarized in Table 1. A forward primer oLLY547 contains a restriction enzyme *Bgl* II site and a backward primer oDG33 has a *Not* I site for insertion of the amplified MmuPV1 Bethesda strain genome into a mammalian p3×FLAG-CMV-14 vector (https://www.addgene.org/vector-database/1621/ (accessed on 28 August 2025)). To amplify and insert the amplified MmuPV1 Bethesda strain genome into a prokaryotic pUC19 vector (https://www.addgene.org/50005/ (accessed on 28 August 2025)), both forward primer oLLY542-BamH I and backward primer oLLY543-*Bam*H I were used. The obtained PCR products were gel-purified, digested with the indicated restriction enzymes (New England Biolabs, Ipswich, MA, USA), and ligated by T4 DNA ligase (New England Biolabs, cat. # M0202T) to the p3×FLAG-CMV-14 to create the plasmid pDG7 or ligated to the pUC19 to create the plasmid pDG9. The ligated products were transformed into *E. coli* TOP10-competent cells (Thermo Fisher Scientific, cat. # C404010). The recombinant colonies were selected at LB agar plates with 100 ug/mL Ampicillin (Thermo Fisher Scientific, cat. # 11593027) and screened through restriction enzyme digestion described above. The selected clones were then sequenced by whole plasmid nanopore sequencing (Quintara Biosciences, Cambridge, MA, USA).

## 3. Results

### 3.1. MmuPV1 Outbreak in an Animal Research Facility

Papilloma-bearing mice initially in rooms A1 and A2 and subsequently in room B five months later were identified in a specific-pathogen-free rodent facility, in which experiments using MmuPV1, a virus that can induce papillomas in laboratory mice, had been carried out in another room (room D) under ABSL-2 conditions as recently as three years prior (Figure 1B). The housing locations of the papilloma-bearing mice and the previous studies using MmuPV1 were physically distant from each other within the facility (Figure 1A,B). All symptomatic mice in room A1 and A2 were euthanized and necropsied, and MmuPV1 was subsequently identified from the papilloma-bearing tissue. Subsequent cages of symptomatic mice prompted extensive investigation into the extent of MmuPV1 contamination of the facility. Widespread MmuPV1 contamination was identified, and many positive cages were found, mostly asymptomatic [14]. The virus was eliminated from the facility through a combination of testing, culling, stopping movement, room depopulation, and facility cleaning and decontamination [14].

No MmuPV1 was previously used in any of the other affected rooms outside room D. All affected animals with clinical signs were 5–10-month-old immunodeficient mice and imported from different vendors, where there was no previous history of MmuPV1 infection.

### 3.2. Necropsy Evaluation of Papilloma-Bearing Animals

Female athymic *NMRI-Foxn1^nu^/Foxn1^nu^* (*n* = 4) mice showing signs of papillomatosis were submitted for a full necropsy evaluation. All animals showed the presence of proliferative, exophytic lesions predominantly affecting the face and oral mucosa, but lesions were also observed on a paw (Figure 1C) and tail. Enlarged mandibular lymph nodes were observed in two mice. Tissues, including the muzzle, haired skin, uterus, cervix, vagina, esophagus, stomach (glandular and non-glandular), duodenum, jejunum, ileum, cecum, colon, rectum, liver, gall bladder, were collected, fixed, and routinely processed for H&E staining. Large skin papilloma masses collected were subdivided using sterile forceps and scalpel blades. Parts of the papillomas were frozen in liquid nitrogen for DNA and RNA extraction, and other parts were fixed for tissue sections used further for RNA-FISH and immunohistostaining.

The proliferative lesions observed on the muzzle, tail, and paw were histologically consistent with virus-induced papillomas. One muzzle lesion appeared invasive with evidence of malignant transformation. The H&E staining showed the hyperplastic epidermis composed of epithelial cells that formed papillary projections supported by thin fibrovascular cores (Figure 2A, left panel). Cells had variably distinct cell borders, a moderate amount of eosinophilic cytoplasm, and round to elongate nuclei with finely stippled chromatin and distinct nucleoli. There was moderate anisocytosis and anisokaryosis and increased mitotic figures. The epithelial cells within the stratum spinosum and granulosum were occasionally enlarged with finely granular, amphophilic cytoplasm and clear cytoplasmic vacuoles. There were also eccentric vesiculate or occasionally pyknotic nuclei surrounded by a clear halo and prominent magenta nucleoli, like typically seen in koilocytosis or koilocytotic atypia (zoomed areas in Figure 2A).

Additional pathologies observed in these infected mice included mild pneumonia for one mouse characterized by a dorsoventral distribution of vasculitis, inflammatory infiltrates, and type II pneumocyte hyperplasia. Additional inflammatory lesions were observed in the uterus and colon.

### 3.3. Characterization of MmuPV1 Gene Expression in the Detected Papillomas

To identify whether MmuPV1 infection was causing the observed papillomatosis in the mouse muzzles and tails, serial tissue sections from the mouse papillomas were subjected to RNA-FISH detection using antisense RNAscope probes targeting MmuPV1 early (E6/E7) and late (L1/L2) transcripts, which were successfully used to detect MmuPV1 RNAs in our previous reports [4,13]. As shown in Figure 2B, the selected papilloma tissues exhibited a robust expression of both early and late viral RNAs, with the MmuPV1 early RNAs preferentially in the lower levels of the epidermis and the MmuPV1 late RNAs in the upper levels. This is in agreement with keratinocytes’ differentiation-driven papillomavirus gene expression [15]. The authentic MmuPV1 RNA transcripts were confirmed not to be viral DNA signals because these MmuPV1 RNA signals were resistant to DNase I treatment [4,13]. MmuPV1 late gene expression in the papilloma tissues was further confirmed by L1 immunohistostaining using an L1-specific antibody (Figure 2C). The anticipated L1 protein expression was clearly associated with highly differentiated keratinocytes in the upper layers of the epidermis, which showed a high topological correlation with the expression of L1 RNAs detected by RNA-FISH (Figure 2B).

In conclusion, the papilloma tissues collected from affected animals not only showed typical histopathological changes associated with papillomavirus infection, but the detection of MmuPV1 RNA transcripts and L1 protein further confirmed MmuPV1 as a causative agent responsible for this recurrent papillomatosis outbreak in the animal facility. Subsequently, we named this causative agent as the MmuPV1 Bethesda strain.

### 3.4. Tracing the Origin of Outbreak MmuPV1

To trace the origin of the MmuPV1 Bethesda strain causing the recurrent papillomatosis outbreak in the animal facility, the extracted total DNA from a representative muzzle papilloma tissue was examined via five separate overlapping PCR reactions with the primer pairs spanning the entire MmuPV1 genome (Figure 3A,B, Table 1). As shown in Figure 3C, all MmuPV1 fragments were equally amplified with the predicted sizes as an indication of the presence of the episomal MmuPV1 genome. The obtained PCR fragments were then gel-purified and subjected to Sanger sequencing. The obtained MmuPV1 Bethesda strain genome sequence from this papilloma tissue was then compared to three other available MmuPV1 sequences: the synthetic reference genome (GenBank Accession No. NC_014326) used by the group that originally studied MmuPV1 infections in this vivarium [10], which was based on published sequence GenBank Accession No. GU808564) [16], the Wild-type (WT) MmuPV1 genome (GenBank Acc. No. PX245442) that was independently isolated by Paul Lambert’s laboratory at the University of Wisconsin-Madison from papilloma extract provided by Dr. Aravind Ingle (ACTREC, India) from the original colony of nude mice infected with MmuPV1 [17], and the MmuPV1 German variant (GenBank Acc. No. HQ625439) [18]. The sequence alignment analysis showed that the genome sequence of the MmuPV1 Bethesda strain (GenBank Accession No. PX123224) is the same as the synthetic MmuPV1 genome (NC_014326) used in the studies previously carried out in room D in this vivarium [10,19], including all three nucleotide variations (one silent mutation in the E1 and two missense mutations in L2 ORFs), which are distinguishable from the WT sequence (GenBank Acc. No. PX245442) (Figure 3D). A T-to-C at nt 2253 leads to no (silent) amino acid change (Y504Y) in the E1 protein, a T-to-A at nt 4499 results in an amino acid change or missense mutation (V252E) in the L2 protein, and a T-to-G at nt 5069 also leads to an amino acid change (L442R) in the L2 protein. Interestingly, at these nucleotide positions, the German variant HQ625439 shares sequence with both synthetic and Bethesda strain virus at the positions 2253 and 5069, but at position 4499, the German variant only shares a sequence with the PX245442 genome (Figure 3D). We observed numerous additional variations between the German variant and WT, synthetic and Bethesda strain (Appendix A). The same MmuPV1 Bethesda strain genome sequence was also confirmed from the papilloma tissues of five other animals. Together, these data show that the obtained Bethesda strain virus from the outbreak had originated from environmental contamination by the synthetic MmuPV1 virus used in previous MmuPV1 experiments carried out in the affected animal facility.

### 3.5. Cloning of the Outbreak MmuPV1 Genome

To clone the MmuPV1 Bethesda strain virus from the affected animals in this outbreak, we amplified the entire viral genome from the collected papilloma tissues using primers (Pr) containing unique restriction enzyme cutting sites (E) and cloned into two corresponding plasmid vectors (Figure 4A). Two distinct plasmids containing the MmuPV1 Bethesda strain genome were generated: pDG7, which has the MmuPV1 genome cloned between *Bgl* II and *Not* I sites of a eukaryotic expression vector p3×FLAG-CMV (Figure 4B), and pDG9, which has the MmuPV1 genome cloned into a prokaryotic vector pUC19 using *Bam*H I site naturally occurring in the viral genome (Figure 4C). In the pGD7 plasmid, the viral E6 ORF is located immediately downstream of an immediate early cytomegalovirus promoter (P_cmv_) in a sense orientation, allowing for substantial expression of viral RNAs in most mammalian cells. In the pDG9 plasmid, the full-length MmuPV1 genome can be retrieved from the plasmid pDG9 DNA by *Bam*H I digestion to delete the pUC19 sequence, and after re-ligation of purified viral DNA, a replication competent episomal viral genome could be produced for MmuPV1 transfection or infection of a susceptible cell or host.

## 4. Discussion

Animal papillomavirus infections are common. To date, more than 200 species-specific animal papillomavirus types have been reported (accessed on 28 August 2025, https://pave.niaid.nih.gov). While other animal viruses have historically been used to study papillomavirus infection, MmuPV1 has gained popularity due to its ability to infect laboratory mice, a well-defined host allowing a wide range of genetic and therapeutic manipulations. Therefore, MmuPV1 recently became a primary experimental surrogate for understanding human papillomavirus biology and pathogenesis [7,11,12]. There are publications outlining standard experimental approaches, including using the synthetic MmuPV1 DNA to infect scarified mouse skin or wounded mucosa. This has allowed numerous studies to be performed and compared to evaluate host susceptibility with different genetic backgrounds, natural history, transmission, immune responses, carcinogenesis, therapeutic treatment, vaccination, etc. However, despite this effort, no standardized biocontainment or clinical care guidelines are available today for the scientific community or research facilities to safely conduct MmuPV1 research in experimental animals.

In this report, we described an unanticipated recurrent MmuPV1 papillomatosis outbreak in an animal facility. Numerous visible skin lesions characteristic of MmuPV1 infection at various animal anatomical sites were observed in immunocompromised mice highly susceptible to MmuPV1 infection [19,20]. Histological evaluation confirmed the cytological changes associated with papillomavirus infection, and active MmuPV1 replication was confirmed by detection of MmuPV1 early and late gene expression. All examined papilloma lesions were positive for MmuPV1 DNA. We ultimately cloned the causative MmuPV1 genome causing the papillomatosis outbreak in the animal facility and designated it as the MmuPV1 Bethesda strain.

The molecular analysis of the MmuPV1 Bethesda strain genome sequences revealed its origination from the synthetic MmuPV1 genome (reference sequence NC_014326) [9,10,16], based on the published GenBank sequence Acc. No.GU808564 [16]. The synthetic MmuPV1 genomic DNA was infectious and papilloma-inducible in athymic nude mice [9,10] and, thus, has been widely distributed to many laboratories around the world. Both the NC_014326 and the Bethesda strain sequences (GenBank Acc. No. PX123224) differ from the WT MmuPV1 genomic DNA (GenBank Accession No. PX245442) [17], which was reisolated from the tumor tissues obtained from the originally reported study, by three nucleotide substitutions located in the E1 and L2 coding regions, with one silent mutation in the E1 ORF and two missense mutations in the L2 ORF (Figure 3D). One possibility is that these three mutations were introduced during viral genome amplification and sequencing. The GU808564-derived reference sequence NC_014326 was obtained by sequencing viral DNA amplified from the tumor tissues by rolling cycle amplification (RCA) using high-fidelity Phi29 DNA polymerase [16], whereas the tumor tissues from the original colony of nude mice infected with MmuPV1 was used to extract DNA and sequenced again with a newer sequencing platform in the Paul Lambert Laboratory [17]. However, the two independent isolates (GU808564 and PX245442) are WT MmuPV1 virus and are capable to induce a productive infection in immunocompromised mice. The second possibility for these observed variations is the naturally occurring variability in the MmuPV1 genome from individual isolates, as seen in HPV variants with intratypic sequence variations [21,22,23]. Interestingly, the same T at nt 2253 and nt 5066 position in the synthetic genome (NC_014326) is also presented in the German variant genome (GenBank Acc. No. HQ625439) [18], but the A at nt 5312 position in the WT PX245442 genome and the German variant genomes is not in the NC_014326 genome. These data indicate the observed sequence differences between the WT PX245442 genome and German variant are naturally occurring variations in MmuPV1 isolates, rather than amplification artifacts. The MmuPV1 German variant genome additionally has more than 60 substitutions, of which a third are missense mutations, and 13 additional nucleotide insertions, of which 3 are inserted within the E2/E4 coding region, making these two proteins one amino acid residue longer than the expected size, and 10 are inserted in the 3ʹUTR of L1 mRNA (Appendix A). These changes indicate the presence of substantial geographical variations in MmuPV1 genomes. Similar differences were observed in CRPV genomes of different origins [24].

The unexpected recurrent MmuPV1 outbreak in a specific pathogen-free animal facility, caused by the virus used in studies conducted three years earlier, was surprising, as no similar cases have been reported in the literature. Our observation and systemic investigation suggest that MmuPV1 can survive and persist in a contaminated environment for extended periods, raising important implications for biosecurity and facility decontamination protocols. This is consistent with previous research demonstrating the high stability of MmuPV1 in the cornified layer and its resistance to many widely used disinfectants and cleaning solutions [25,26,27]. We suspect that the introduction of immunocompromised mouse strains commonly used in cancer and immunology research has provided highly susceptible hosts, resulting in an uncontrolled MmuPV1 infection and spreading a large amount of virus into the environment across the facility. Various strains of mice were infected during this outbreak, most without clinical signs [14]. This raises a serious concern, as undetected MmuPV1 infection may alter experimental outcomes and lead to misinterpretation of ongoing studies, potentially causing substantial scientific and economic harm. Furthermore, as the origin of the first reported MmuPV1 infection at the Advanced Centre for Treatment Research and Education in Cancer (ACTREC) in Mumbai, India, has never been sufficiently explained, we speculate that the MmuPV1 infections in the ACTREC might be caused by locally circulating MmuPV1 introduced to the facility from wild animals, contaminated clothing, or contaminated materials such as bedding or food. This argues for the possibility that MmuPV1 outbreaks will also occur in the future in facilities without a history of experimental use of MmuPV1.

Therefore, our study strongly advocates the development of strict guidelines for the experimental use of MmuPV1, including proper prevention, monitoring and elimination of unintended spread of MmuPV1 infection among laboratory animals. The naturally occurring nucleotide substitutions in the MmuPV1 genome described in this report may represent a traceable molecular marker to identify the source of the infection.

## Figures and Tables

**Figure 1 viruses-17-01204-f001:**
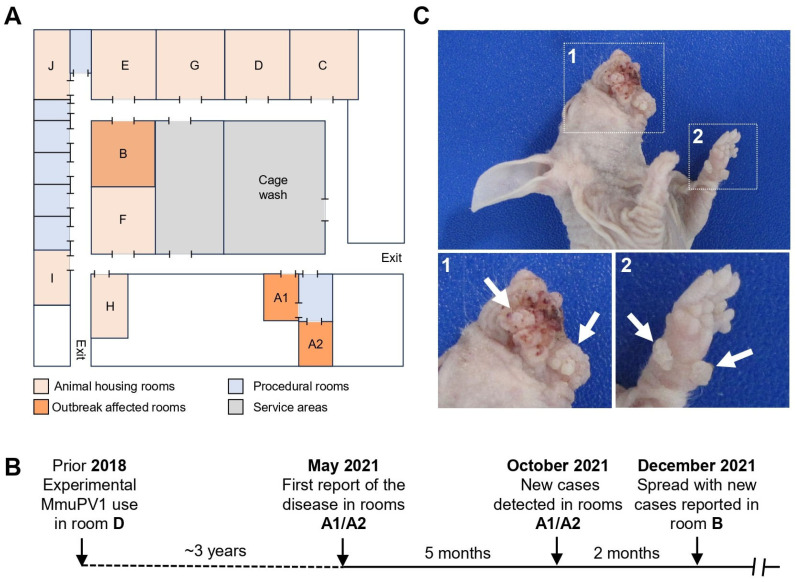
Natural history of recurrent papillomatosis outbreak in animal research facility. (**A**) The floor plan of the affected facility showing the affected rooms. (**B**) Timeline of the noticed papillomatosis in the facility. (**C**) The representative papilloma-resembling nodules noticed on the muzzle (**upper panel**) and paw (**lower panel**) of one of the affected animals.

**Figure 2 viruses-17-01204-f002:**
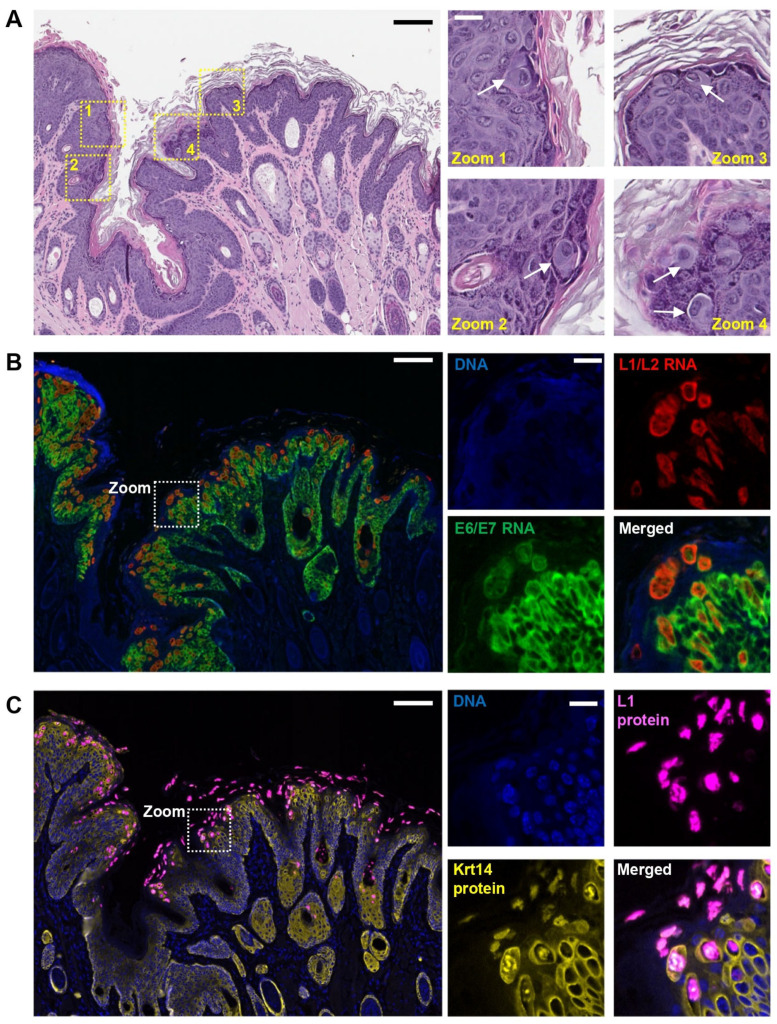
Detection of MmuPV1 expression in the muzzle skin papilloma. (**A**) Papilloma histopathology. H&E staining of papilloma section from the muzzle skin collected from a female mouse. Showing on the right are four zoomed tissue section areas containing koilocytes (white arrow) typical for papillomavirus infection. (**B**) Expression of MmuPV1 early (E6/E7, green) and late (L1/L2, red) transcripts from the muzzle skin papilloma (**A**) detected by dual immunofluorescent RNA in-situ hybridization (RNA-ISH) by RNAscope with an antisense probe to E6/E7 (green) and an antisense probe to L1/L2 (red). The cell nuclei stained by DAPI are shown in blue. The signal intensities for the individual channels in the zoomed area are shown on the right. (**C**) The protein expression of MmuPV1 L1 (pink) and mouse Krt14 (yellow) from the muzzle skin papilloma sections. The tissue sections in (**A**) were immunostained by dual immunofluorescence staining with anti-MmuPV1 L1 as previously described [13] and anti-mouse Krt14 antibodies. The nuclei stained by DAPI are shown in blue. The detected L1 and Krt14 protein staining in the zoomed area shown on the right. Scale bars equal 100 μm in the left panel and 25 μm in the right zoomed panels.

**Figure 3 viruses-17-01204-f003:**
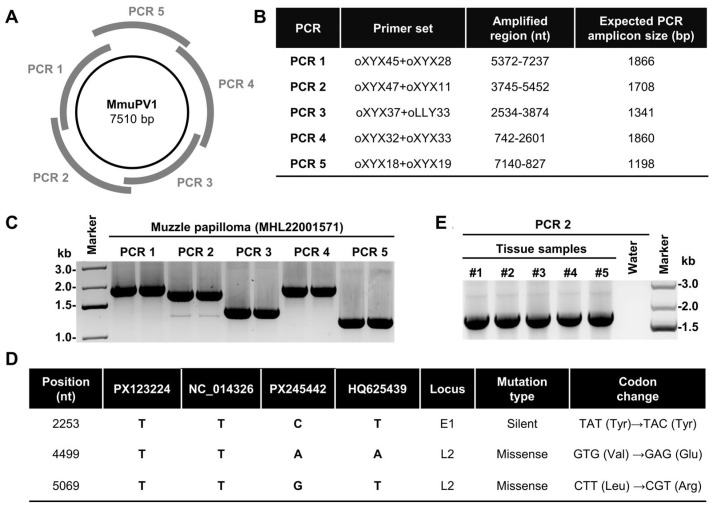
Sequence analysis of the MmuPV1 Bethesda strain genome. (**A**) The diagram of the MmuPV1 genome with location of five overlapping PCR amplicons (PCR 1-5) designed to cover the entire viral genome using primers shown in (**B**). The nt positions and the expected size of PCR products are based on the published MmuPV1 transcription map [13]. (**C**) Amplification of the entire MmuPV1 genome from a representative Bethesda strain-induced papilloma. The individual DNA fragments were gel purified and sequenced by Sanger sequencing. (**D**) The comparison of MmuPV1 Bethesda strain (PX123224) genome sequence with the genome sequences of the synthetic MmuPV1 (NC_014326) [16], wild-type (WT, PX245442) [17], and MmuPV1 German variant (HQ625439) [18] exhibiting three nt variations in the E1 and L2 ORFs. (**E**) The amplified MmuPV1 DNA products in multiple papilloma tissues collected from additional infected animals exhibit the same MmuPV1 sequence of the Bethesda strain.

**Figure 4 viruses-17-01204-f004:**
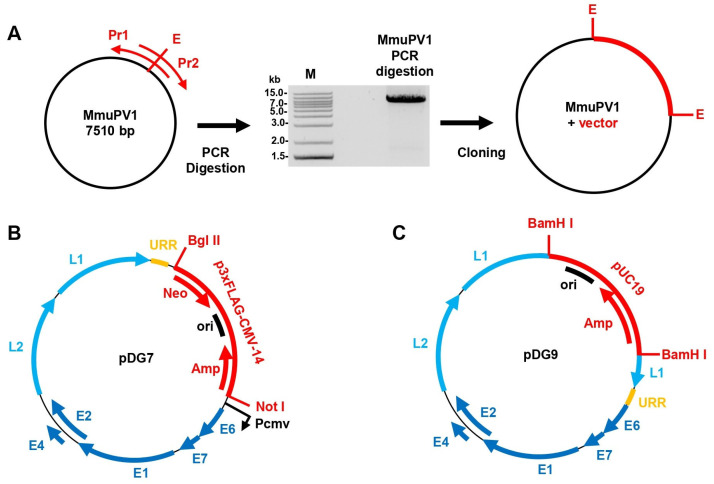
Cloning of MmuPV1 Bethesda strain genome. (**A**) The MmuPV1 genome cloning strategies. The full-length viral genome (black circle) was amplified by PCR from total DNA extracted from the Bethesda strain-induced mouse papilloma using two overlapping primers (red arrows) containing restriction enzyme sites (oLLY547-*Bgl*II and oDG33-*Not*I for plasmid pDG7; oLLY542-*Bam* HI and oLLY543-*Bam* HI for plasmid pDG9). The obtained PCR products were digested with the indicated restriction enzymes and ligated with a vector p3×FLAG-CMV-14 to create the plasmid pDG7 (**B**) or ligated with pUC19 to create the plasmid pDG9 (**C**). (**B**) The map of constructed pDG7 plasmid has the MmuPV1 genome of the Bethesda strain cloned into the p3×FLAG-CMV-14 vector at *Not*I and *Bgl*II sites to make a cytomegalovirus immediate early promoter (P_cmv_) upstream of the viral E6 ORF for transfection and MmuPV1 expression in mammalian cells. (**C**) The map of constructed pDG9 has the MmuPV1 genome of the Bethesda strain cloned into a pUC19 vector via a *Bam*HI site for amplification in *E. coli*. The circular viral genome could be re-constituted by re-ligation after *Bam*HI digestion to delete the pUC19 sequence. URR-upstream regulatory region.

**Table 1 viruses-17-01204-t001:** Oligo primers used in this study. All nucleotide positions are based on the MmuPV1 reference genome GenBank Acc. No. NC_014326. URR-upstream regulatory region. N/A: not applicable. Restriction sites are underlined. Slash separates viral and non-viral sequences.

Name	Strand	5′ (nt)	3′ (nt)	Locus	Restriction Site	Sequence
oLLY33	B	3874	3855	L2	N/A	TGTCAGCAAGTGTGTTTCCT
oXYX11	B	5452	5433	L1	N/A	TTCGTCTGTGCTCTGCACTT
oXYX18	F	7140	7160	URR	N/A	TGTTGGCTGTGTGCTCTCTAA
oXYX19	B	827	807	E1	N/A	AAGGAACCCACATCATCCACA
oXYX28	B	7237	7216	URR	N/A	CAAATTGGCTGGAGTTTATGCT
oXYX32	F	742	762	E1	N/A	ATGGAAAACGATAAAGGTACA
oXYX33	B	2601	2582	E1/E2	N/A	CTGCCTTTCTCGTAAAGGTT
oXYX37	F	2534	2554	E1/E2	N/A	ATGAACAGCCTGGAAACACGT
oXYX45	F	5372	5391	L1	N/A	ATGGCAATGTGGACACCCCA
oXYX47	F	3745	3763	L2	N/A	ATGGTGTCTGCTGACAGAA
oLLY547	B	7502	7483	URR	*Bgl*II	ATCATATAGATCT/ACGGTTATGGGGGCACACTG
oDG33	F	7503	12	URR/E6	*Not*I	GCGCATACTGCGGCCGC/ATTCGTTCATGGAAATCGGC
oLLY542	F	6689	6716	L1	*Bam*HI	GAAAGAGAGGATCCTTACAAGGGTCTTA
oLLY543	B	6708	6681	L1	*Bam*HI	TTGTAAGGATCCTCTCTTTCCTTGGGCG

## Data Availability

The original contributions presented in this study are included in the article. Further inquiries can be directed to the corresponding authors. The genomic sequence of MmuPV1 Bethesda strain is accessible at the NCBI GenBank under Accession No. PX123224 and MmuPV1 WT sequence with a GenBank Accession No. PX245442 is accessible at the NCBI GenBank.

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
