# Peer review of "Identification and Characterization of MmuPV1 Causing Papillomatosis Outbreak in an Animal Research Facility"

_viruses, 2025, doi:10.3390/v17091204_

Round 1

Reviewer 1 Report

Comments and Suggestions for Authors

This manuscript describes an unexpected MmuPV1 outbreak in an animal research facility and demonstrates that the source of infection originated from an MmuPV1 experiment conducted at the same facility three years earlier. This finding indicates that MmuPV1 can persist undetected for extended periods in a contaminated environment and highlights the inadequacy of current SOPs for animal experiments involving MmuPV1. If presented together with the data, currently in submission, showing that asymptomatic infection has also been observed in other immunocompetent mice, the work could become an even more impactful and important report; however, it may also be considered beyond the scope of the current manuscript, and the present study alone offers substantial value to the readership.

Major/Minor

Although noted as “in submission,” (i) the method of disinfection/decontamination and how its effectiveness was verified, and (ii) the observation of asymptomatic infection in other mice, are both closely related to the content of this manuscript. If these points are to be mentioned, they should ideally be reported within the same paper. If the other study has been submitted to the same journal (Viruses), it should be treated as an accompanying manuscript. If that manuscript is accepted prior to the publication of the present study, the relevant descriptions in this manuscript should be updated accordingly.

line 47 :However, given the wide range of spe-46 cies-specific viruses, other animal papillomaviruses have been used "in what". Please clarify.

Section 3.5 reads more like a “Materials and Methods” section than part of the “Results.” It might be appropriate to consider moving the methodological details to the appropriate section.

Author Response

Reviewer #1 Comments and Suggestions for Authors

This manuscript describes an unexpected MmuPV1 outbreak in an animal research facility and demonstrates that the source of infection originated from an MmuPV1 experiment conducted at the same facility three years earlier. This finding indicates that MmuPV1 can persist undetected for extended periods in a contaminated environment and highlights the inadequacy of current SOPs for animal experiments involving MmuPV1. If presented together with the data, currently in submission, showing that asymptomatic infection has also been observed in other immunocompetent mice, the work could become an even more impactful and important report; however, it may also be considered beyond the scope of the current manuscript, and the present study alone offers substantial value to the readership.

Major/Minor

1. Although noted as “in submission,” (i) the method of disinfection/decontamination and how its effectiveness was verified, and (ii) the observation of asymptomatic infection in other mice, are both closely related to the content of this manuscript. If these points are to be mentioned, they should ideally be reported within the same paper. If the other study has been submitted to the same journal (Viruses), it should be treated as an accompanying manuscript. If that manuscript is accepted prior to the publication of the present study, the relevant descriptions in this manuscript should be updated accordingly.

Thank you for the comments. The manuscript noted as “in submission” has been formally accepted for publication by the Journal of the American Association for Laboratory Animal Science (JAALAS) and is cited in the revised manuscript.

2. line 47: However, given the wide range of species-specific viruses, other animal papillomaviruses have been used "in what"? Please clarify.

We apologize for the incomplete statement, which was corrected in the revised text as follows: “However, given the wide range of species-specific viruses, other animal papillomaviruses have been used as HPV surrogates”.

3. Section 3.5 reads more like a “Materials and Methods” section than part of the “Results.” It might be appropriate to consider moving the methodological details to the appropriate section.

Thanks for your suggestion. We would like to keep the current description as a result, which outlined the strategies and cloning result. More details are described in the Materials and Methods.   

Reviewer 2 Report

Comments and Suggestions for Authors

This paper presents a fascinating and concerning account of a viral reinfection. The study reports the seemingly unbelievable fact that a virus, which was last used in an advanced research center three years prior, caused a recurrent outbreak despite the facility having robust biosafety and decontamination protocols.

The findings have serious implications, as they could potentially undermine the reliability and integrity of many other studies conducted at the same location. Above all, this research is significant enough for publication because it effectively raises the critical issue of developing standardized biocontainment and clinical care guidelines.

Overall, the manuscript is well-written and organized. The authors clearly present the results from the virus identification and characterization process. However, because the study does not report the discovery of a new viral strain, there is a question regarding the originality and novelty of the characterization research.

The primary classification of this virus, which is typically performed by L1 PCR sequencing, would have resulted in its classification as the same virus as the old one. However, a full genome analysis revealed three nucleotide mutations (one silent mutation and two missense mutations in L2). It is suggested that the authors submit their findings for review by the ICTV (International Committee on Taxonomy of Viruses) to determine if these mutations are sufficient for it to be designated as a new strain. The reviewer is curious whether naming it the Bethesda isolate would be a more appropriate designation.

Author Response

Reviewer #2 Comments and Suggestions for Authors:

This paper presents a fascinating and concerning account of a viral reinfection. The study reports the seemingly unbelievable fact that a virus, which was last used in an advanced research center three years prior, caused a recurrent outbreak despite the facility having robust biosafety and decontamination protocols.

1. The findings have serious implications, as they could potentially undermine the reliability and integrity of many other studies conducted at the same location. Above all, this research is significant enough for publication because it effectively raises the critical issue of developing standardized biocontainment and clinical care guidelines.

We thank the reviewer for recognizing the importance of the presented work.

 2. Overall, the manuscript is well-written and organized. The authors clearly present the results from the virus identification and characterization process. However, because the study does not report the discovery of a new viral strain, there is a question regarding the originality and novelty of the characterization research.

Thank you for the comments. We believe our work poses important value that MmuPV1 can persist undetected for extended periods in a contaminated environment and highlights the inadequacy of current SOPs for animal experiments involving MmuPV1. In addition, our results provide compelling evidence that the infection and outbreak originated from the laboratory synthetic viral genomic DNA. Thus, our study showed the first isolation of MmuPV1 from infected animals, resulting in its name as the Bethesda strain.

3. The primary classification of this virus, which is typically performed by L1 PCR sequencing, would have resulted in its classification as the same virus as the old one. However, a full genome analysis revealed three nucleotide mutations (one silent mutation and two missense mutations in L2). It is suggested that the authors submit their findings for review by the ICTV (International Committee on Taxonomy of Viruses) to determine if these mutations are sufficient for it to be designated as a new strain. The reviewer is curious whether naming it the Bethesda isolate would be a more appropriate designation.

Yes. The classification of papillomavirus was typically carried out by L1 PCR sequencing in the past as the whole-genome sequencing was difficult then. Our study in this report presents the first isolation of MmuPV1 in an outbreak in a research animal facility. As there is no MmuPV1 report from naturally infected mice in North America, our isolated MmuPV1 in outbreak happened to a population of a group of mice repeatedly in the research facility. We prefer to name it as a Bethesda strain, NOT a viral isolate from a single animal or a few animals, despite its origination from a synthetic viral genome used for previous studies in the facility. According to Van Regenmortel, a (natural) virus strain is a “variant of a given virus that is recognizable because it possesses some unique phenotypic characteristics that remain stable under natural conditions (PMID: 16713373).   

Reviewer 3 Report

Comments and Suggestions for Authors

This is an important study that warrants publication, but several key details are missing that are necessary to fully understand the spread of infection:

  1. What decontamination protocols were originally performed in Room D?

  2. Which immunocompromised mouse strains were infected in Rooms A and B, and from which vendors were they from?

  3. Did the animals that developed warts or disease exhibit any aggressive behavior or fighting that could have increased wounding and susceptibility?

  4. What are the current animal care protocols in Rooms A and B, and were the same protocols originally used in Room D?

  5. When disease was first observed in Room A, were the affected mice euthanized immediately? If not, how long did they remain alive with visible disease?

  6. The authors note that other facilities have not reported outbreaks. What specific protocol changes do they recommend to prevent the spread of MmuPV1?

  7. Following this outbreak, what changes—if any—have been implemented in the animal facility to limit the spread of MmuPV1 or other pathogens?

Author Response

Reviewer 3 Comments and Suggestions for Authors:

This is an important study that warrants publication, but several key details are missing that are necessary to fully understand the spread of infection:

Thank you for your excellent questions.  The answers to these questions have been addressed in a separate manuscript, which has just been accepted for publication in the Journal of the American Association for Laboratory Animal Science.  The manuscript (Killoran KE, Breed MW, Roelke-Parker ME, Carney S, Edmondson E, Thompson CD, Schiller JT, Henderson K, Woods CL, Albers TM, Starost MF, and Kramer JA. Mouse Papillomavirus outbreak in a research facility. JAALAS, accepted) details the discovery of a small group of clinically infected Foxn1nu/nu mice, identification of additional Nude mice from two vendors suggesting in-facility transmission, diagnosis of infection, development of a diagnostic PCR assay, assessment of the extent of infection throughout the facility, surveillance and elimination of infected mice from the facility, and cleaning and disinfection procedures that were undertaken.

1. What decontamination protocols were originally performed in Room D?

Cleaning and disinfection in Room D prior to the pandemic were performed with sodium hypochlorite. All cages were handled in the biosafety cabinet. All waste, including carcasses, was handled as medical pathological waste, which is incinerated. 

2. Which immunocompromised mouse strains were infected in Rooms A and B, and from which vendors were they from?

All clinically affected mice were exclusively of the Foxn1nu/nu genotype, less than one year of age, and mostly female. The initial cases were observed in animals purchased from Charles River Laboratories, and other animals were from Taconic. The extent of disease varied among individuals. Some animals had small lesions on the mouth onl,y and others had extensive lesions on the mouth, paws, and tail.

3. Did the animals that developed warts or disease exhibit any aggressive behavior or fighting that could have increased wounding and susceptibility?

There were no reports of aggression, such as health reports for fighting animals, in any of the cages.

4. What are the current animal care protocols in Rooms A and B, and were the same protocols originally used in Room D?

Current animal care protocols are the same in Rooms A, B, and D, with all cages being handled in the biosafety cabinet and all waste handled as medical pathological waste.  Aseptic cage changes occur in all three rooms.

5. When the disease was first observed in Room A, were the affected mice euthanized immediately? If not, how long did they remain alive with visible disease?

Once identified with lesions, all clinically affected animals were euthanized. Necropsy for some animals was arranged no more than a week.

6. The authors note that other facilities have not reported outbreaks. What specific protocol changes do they recommend to prevent the spread of MmuPV1?

To prevent outbreaks of MmuPV1, we suggest that animals be housed in ABSL-3 level due to the virus stability and its risk that it poses to the rest of the facility.  Our ABSL-2 room is within the corridor with other animals of higher health status, but there is no mechanism to prevent investigators from breaking room order or using contaminated supplies in other rooms, other than the rules within the facility.  A highly access-restricted facility is more appropriate for housing animals that are infected with MmuPV1.

7. Following this outbreak, what changes—if any—have been implemented in the animal facility to limit the spread of MmuPV1 or other pathogens?

The facility will no longer allow MmuPV1 to be used within any room.  The investigator who originally used the virus has also said that he will never work with it again.  If we were to allow its use again, we would require the virus to be used in a contracted space that is ABSL-3.  We have not, to our knowledge, had any other pathogens that are used in the ABSL-2 room escape that room.  However, we have expanded our sentinel program to include more comprehensive testing than in the past.  We have also become far stricter with the users of the facility.  Unfortunately, this outbreak impacted every user in the facility, and compliance has been quite good because no one wants to be responsible for setting their own or others’ research back due to the need to depopulate rooms.